# Survival Evidence of Local Control for Colorectal Cancer Liver Metastases by Hepatectomy and/or Radiofrequency Ablation

**DOI:** 10.3390/cancers15184434

**Published:** 2023-09-06

**Authors:** Lariza Marie Canseco, Yueh-Wei Liu, Chien-Chang Lu, Ko-Chao Lee, Hong-Hwa Chen, Wan-Hsiang Hu, Kai-Lung Tsai, Yao-Hsu Yang, Chih-Chi Wang, Chao-Hung Hung

**Affiliations:** 1Section of Gastroenterology, Department of Internal Medicine, De Los Santos Medical Center, Quezon City 1112, MM, Philippines; lariza.canseco@gmail.com; 2Liver Transplant Center, Department of Surgery, Kaohsiung Chang Gung Memorial Hospital and Chang Gung University College of Medicine, Kaohsiung 833401, Taiwan; anthony0612@adm.cgmh.org.tw (Y.-W.L.); chihchiwang@cgmh.org.tw (C.-C.W.); 3Department of Colorectal Surgery, Kaohsiung Chang Gung Memorial Hospital and Chang Gung University College of Medicine, Kaohsiung 833401, Taiwan; lu019@gmail.com (C.-C.L.); kclee@cgmh.org.tw (K.-C.L.); kmch4329@gmail.com (H.-H.C.); gary.hu0805@gmail.com (W.-H.H.); kltsai@cgmh.org.tw (K.-L.T.); 4Department of Traditional Chinese Medicine, Chiayi Chang Gung Memorial Hospital, Chiayi 61363, Taiwan; r95841012@cgmh.org.tw; 5Health Information and Epidemiology Laboratory of Chang Gung Memorial Hospital, Chiayi 61363, Taiwan; 6Division of Hepatogastroenterology, Department of Internal Medicine, Kaohsiung Chang Gung Memorial Hospital and Chang Gung University College of Medicine, Kaohsiung 833401, Taiwan

**Keywords:** colorectal cancer, liver metastases, hepatic resection, radiofrequency ablation, overall survival

## Abstract

**Simple Summary:**

Colorectal cancer with liver metastases (CRLM) has a poor prognosis. Systemic treatment alone, or worse, best supportive care, only affords patients limited survival. This study aims to provide evidence that aggressive local control through hepatic resection and/or radiofrequency ablation (RFA) can significantly prolong CRLM overall survival. Out of the 2612 patients enrolled in this study, 637 underwent hepatectomy, 93 had RFA, 92 were given combined hepatectomy and RFA, while 1790 received non-aggressive treatment. Based on the Kaplan–Meier curves and multivariate Cox’s regression analysis as well as frequency matching analysis, we conclude that aggressive local control in CRLM patients has survival benefits, in addition to systemic therapy from a large multi-institutional database.

**Abstract:**

Hepatectomy and/or local ablation therapy have been recommended for colorectal cancer liver metastases (CRLM). However, they still lack strong evidence for their survival benefits, in addition to systemic therapy. This study aims to evaluate the survival evidence of hepatectomy and/or radiofrequency ablation (RFA) therapy in CRLM patients from a large multi-institutional database. A total of 20,251 patients with colorectal cancer, 4521 of whom were with CRLM, were screened for eligibility. Finally, 2612 patients (637 hepatectomy, 93 RFA, 92 combined hepatectomy and RFA, and 1790 non-aggressive treatment) were enrolled. Frequency matching analysis was used to adjust for baseline differences. The 5-year overall survival (OS) was as follows: hepatectomy alone was 47.8%, combined hepatectomy plus RFA was 35.9%, RFA alone was 29.2%, and the non-aggressive treatment group was 7.4%. Kaplan–Meier curves showed that hepatectomy, RFA, and combination were significantly associated with a better OS compared to those without aggressive local therapy (*p* < 0.001). Multivariate Cox regression analysis showed that male gender (hazard ratio (HR) 0.89; 95% confidence interval (CI), 0.81–0.97; *p* = 0.011), old age (≥60 years) (HR 1.20; 95% CI, 1.09–1.32; *p* < 0.001), high CEA level (>5 ng/mL) (HR 2.14; 95% CI, 1.89–2.42; *p* < 0.001), primary right-sided cancer (HR 1.35; 95% CI, 1.22–1.51; *p* < 0.001), extrahepatic metastasis (HR 1.46; 95% CI, 1.33–1.60; *p* < 0.001), systemic therapy (HR 0.7; 95% CI, 0.62–0.79; *p* < 0.001), and aggressive local therapy (hepatectomy vs. non-local therapy HR 0.22; 95% CI, 0.20–0.26; *p* < 0.001; RFA vs. non-local therapy HR 0.29; 95% CI, 0.29–0.41; *p* < 0.001) were independent factors associated with OS. In the frequency matching analysis, patients receiving hepatectomy and/or RFA resulted in a better OS than those without (*p* < 0.001). In conclusion, aggressive local treatment provides survival advantages over systemic therapy alone among CRLM patients.

## 1. Introduction

Colorectal cancer (CRC) remains the second deadliest cancer in the world with 935,173 mortalities, and is still the third highest incidence of cancer with over 1.9 million new cases reported in 2020 [1]. Asia appears as the top contributor of CRC, accounting for around half of all new cases (52.3%), mortalities (54.2%), and 5-year prevalence (49.9%) [1]. The stage upon the diagnosis of CRC is a crucial prognostic factor for survival. Data from the American Cancer Society show that the 5-year survival of CRC with localized-stage disease is 90% [2]. This decreases to 71% for patients with regional metastases and with a subsequent significant drop among cases with distant metastases to 14% [2]. It is estimated that around 20% to 50% of CRC patients will develop metastases along the course of their illness [3]. A population-based study in Sweden noted that the liver was the most frequent site for metastases at 26.5%, followed by the lungs (16.9%), peritoneum (7.1%), and distant lymph nodes (4.8%), which were similar to previously reported trends [3,4]. Synchronous colorectal cancer liver metastases (CRLM) were found to occur in 14.5 to 56.4% of CRC cases, while reports of metachronous lesions were about 10.3–19.6%, with the majority occurring at higher TNM stages on initial diagnosis [3,5,6,7,8].

Even though surgery is primarily considered the standard approach for the curative treatment of CRLM, only 6.1 to 25.4% eventually undergo hepatic metastasectomy [3,5,7]. Ideally, R0 resection should be achieved, regardless of whether the surgery is conducted as curative or palliative treatment. In a study by Park et al., a palliative resection for metastatic colorectal cancer with a negative margin (R0) showed significantly longer survival compared to patients with either a positive margin (R1) or grossly residual tumor (R2) (51.3 months versus 19.1 months), and those without resection (14.1 months) [9].

Thermal ablation is a less invasive procedure that has the potential to provide curative intent, as an alternative to resection. This is especially beneficial for patients with unresectable CRLM, patients with co-morbidities preventing them from undergoing surgery, or those with an insufficient liver reserve [10]. Although shorter progression-free and disease-free survivals have been observed among patients who underwent thermal ablation compared to resection, some studies showed no statistical difference when the tumors were less than 3 cm in size [11,12]. The consensus formed by an expert panel recommended tumors less than 3 cm as the preferred size for thermal ablation, but distinct tumors < 5 cm may also have good thermal ablation outcomes, depending on their location and the ablation method used [10].

Several studies have shown increased survival for CRLM patients who underwent resection and/or ablation. However, only surgical resection has been consistently recommended for curative intent [3,13,14,15]. The role of thermal ablation alone or in combination with resection, for local control, is still not well established, and currently lacks strong, uniform recommendations across all guidelines. Of importance, although hepatic resection appears to offer a potential cure and long-term survival rate, there may have been selection bias in previous studies. Until now, there is no direct evidence of control trials to prove the survival benefits of local control for CRLM by hepatectomy and/or radiofrequency ablation (RFA) in addition to systemic therapy.

We therefore conducted a retrospective cohort study using a de-identified database derived from a multi-institutional electronic medical records collection in Taiwan [16]. The survival outcomes of those who underwent resection, RFA, combined resection and RFA, or received systemic therapy alone were compared. We used a statistically valid frequency matching analysis to adjust for baseline differences to compare the overall survival (OS) between patients with hepatectomy and/or RFA and those without.

## 2. Materials and Methods

### 2.1. Patient Selection and Study Design

Between January 2004 and December 2017, a total of 20,251 patients with CRC were screened for eligibility from the Chang Gung Memorial Hospital (CGMH) system. CGMH is the largest medical care system in Taiwan, consisting of 4 tertiary-care medical centers and 3 major teaching hospitals. This medical care system provides nearly 10% of the medical service used by the Taiwanese people annually, with more than 10,000 beds and over 280,000 inpatients per year [16]. Of these cases, 4521 with CRLM were identified. Fifty-three patients who previously received hepatectomy or RFA and 1856 patients with 3 more missing data of baseline characteristics, such as serum carcinoembryonic antigen (CEA) levels or body mass index (BMI), were excluded. Finally, 2612 patients (637 of whom underwent hepatectomy, 93 RFA, 92 hepatectomy combined with RFA, and 1790 non-aggressive treatment) were enrolled. The non-aggressive treatment group were patients who either received systemic therapy alone or best supportive care. The patient selection flowchart is shown in Figure 1.

The baseline characteristics gathered were as follows: gender, age, Diabetes Mellitus, BMI, CEA, primary cancer site (left or right side), clinical stage, timing of liver metastasis (synchronous or metachronous), extrahepatic metastasis, systemic therapy, neo-adjuvant therapy, and adjuvant therapy. Synchronous CRLM were defined as metastatic liver lesions found within 30 days of the primary diagnosis. Neoadjuvant or adjuvant systemic therapy was considered as treatment within 60 days before or after hepatectomy/RFA, respectively. The regimens of systemic therapy included the current standard treatment for CRLM. The OS was defined as the time interval from the date of primary diagnosis to death from any cause or the last follow-up date.

Pre-operative imaging studies included the triphasic enhanced *Computed Tomography* (CT) scan, Magnetic Resonance Imaging (MRI), or the Positron Emission Tomography (PET) scan in doubtful cases. These cases were discussed in a multi-disciplinary meeting to recommend the procedure precedence. The study was approved by the Research Ethics Committee of Chang Gung Memorial Hospital and was conducted in accordance with the principles of Declaration of Helsinki and the International Conference on Harmonization for Good Clinical Practice.

### 2.2. Frequency Matching Analysis

Frequency matching analysis was used to adjust for baseline differences between patients with hepatectomy and/or RFA and those who received non-aggressive treatment at a ratio of 1:1. The data were obtained by systematically conducting simple randomization sampling with frequency matching by age, gender, liver metastases timing, extrahepatic metastasis, and systemic therapy. Overall, 1194 patients (597 matched sets) were included in the matched cohort.

### 2.3. Statistical Analysis

The median survival time as well as 5-year OS were computed for all treatment groups. Kaplan–Meier curves were generated for overall survival and the differences between groups were compared using the log-rank test. Cox proportional hazards models were used to compute the hazard ratios (HRs) accompanying the 95% confidence interval (CI) after adjustment for potential confounders. All of these analyses were carried out using SAS statistical software (Version 9.4; SAS Institute, Cary, NC, USA), and two-tailed *p* < 0.05 was considered to be significant.

## 3. Results

### 3.1. Patient Characteristics

The baseline characteristics of the study population prior to frequency matching analysis are shown in Table 1. There were significant differences among the groups in terms of age distribution, BMI, serum CEA levels, primary cancer site, clinical staging, extrahepatic metastasis, and systemic therapy. Left-sided CRC and clinical stage IV on initial diagnosis were predominant for all treatment groups. Compared to the local control treatment groups, majority (56%) of the non-aggressive treatment group had extrahepatic metastases.

### 3.2. Survival Outcome and Its Associated Factors in the Entire Cohort

The median survival time for hepatectomy, combined hepatectomy plus RFA, and RFA alone (54 months, 48 months, and 30 months, respectively) were significantly longer compared to the non-aggressive treatment group (10.8 months). A similar trend was likewise seen for the 5-year OS of hepatectomy, combined hepatectomy plus RFA, and RFA alone (47.8%, 35.9%, and 29.2%, respectively) in contrast to the non-aggressive treatment group with 7.4%. Kaplan–Meier curves generated from the entire cohort showed that hepatectomy, RFA, and combined treatment were significantly associated with better overall survival compared to those who did not receive aggressive local therapy (*p* < 0.0001) (Figure 2). Multivariate Cox regression analysis revealed that male gender (HR 0.89; 95% CI, 0.81–0.97; *p* = 0.011), systemic therapy (HR 0.7; 95% CI, 0.62–0.79; *p* < 0.001), and aggressive local therapy (hepatectomy vs. non-local therapy HR 0.24; 95% CI, 0.21–0.27; *p* < 0.001; RFA vs. non-local therapy HR 0.33; 95% CI, 0.24–0.45; *p* < 0.001; hepatectomy plus RFA vs. non-local therapy HR 0.28; 95% CI, 0.21–0.37; *p* < 0.001) were independent factors associated with a better OS (Table 2). In contrast, old age (≥60 years) (HR 1.20; 95% CI, 1.09–1.32; *p* < 0.001), a high CEA level (>5 ng/mL) (HR 2.14; 95% CI, 1.89–2.42; *p* < 0.001), primary right-sided colon cancer (HR 1.35; 95% CI, 1.22–1.51; *p* < 0.001), metachronous liver metastases (HR 1.26; 95% CI, 1.14–1.39; *p* < 0.001), and extrahepatic metastasis (HR 1.46; 95% CI, 1.33–1.60; *p* < 0.001) portend a worse OS (Table 2).

### 3.3. Survival Outcome in Frequency Matching Cohort

Table 3 shows the comparison of patients receiving hepatectomy and/or RFA and non-aggressive local treatment under the frequency matching analysis (1:1). After frequency matching, both the median survival time and 5-year OS remained significantly higher in the local control treatment groups compared to the non-aggressive treatment group (combined hepatectomy plus RFA was 45.6 months and 35.3%, hepatectomy alone was 54.0 months and 47.5%, RFA alone was 31.2 months and 26%, and non-aggressive treatment group was 14.4 months and 8.3%). Kaplan–Meier curves showed that hepatectomy, RFA, and combined treatment were significantly associated with a better OS than those who did not receive aggressive local therapy (*p* < 0.0001) (Figure 3). In subgroup analyses of risk for OS among CRLM patients, aggressive local control by hepatectomy, RFA, or combined therapy had a significantly better OS compared to those without, irrespective of the differences in age and gender (Table 4).

## 4. Discussion

This present study provides evidence that aggressive local control by hepatectomy and/or RFA in addition to systemic therapy may prolong OS among CRLM patients from a large multi-institutional database in Taiwan. Unlike previous retrospective reports, we used a frequency matching analysis adjusted by age, gender, liver metastases timing, extrahepatic metastasis, and systemic therapy to confirm the survival benefits in patients who received hepatectomy and/or RFA compared to those who did not. In this study, the non-aggressive treatment group had a higher ratio of extrahepatic metastasis compared to the local control treatment groups, suggesting that extrahepatic metastasis is the primary reason why the reference group did not receive surgery or RFA. However, our frequency matching analysis including extrahepatic metastasis in the matching process and showed the definite survival advantage of local control with resection, RFA, or combination, even though the patients had extrahepatic metastasis. Of interest, there were no significant differences in the OS among groups of hepatectomy, RFA, and combined treatment. However, this issue should be further clarified due to the limited cases of RFA and combined treatment.

In our study, we showed that several independent factors were associated with OS. Older age was a negative predictor for OS, which was similar to a previous literature showing that each additional year is accompanied by an incremental 3% rise in risk of mortality [4]. Similarly, elevated CEA levels were found to increase the mortality risk for patients with CRLM [17,18,19]. CEA is even included in the scoring systems that predict early recurrence and poor outcome post-hepatectomy among CRLM patients [20,21]. In particular, our study showed that right-sided CRC signified a poor prognosis consistent with the reports from other authors [4,22,23]. Patients with liver metastases from left-sided CRC had a much longer 5-year OS of 16.6% compared to 4.3% in right-sided CRC. In addition, right-sided CRC cases were found to have a higher T- and N-stage on initial diagnosis in Engstrand et al.’s study [4]. Differences in their embryonic origins as well as later time of symptom manifestation were attributed to the more extensive and higher number, albeit less frequent tumor presentation in right-sided CRC [4,22,23]. Previous publications also revealed a molecular basis for this as well, with right-sided CRC having higher microsatellite instability and BRAF mutations as well as defective DNA mismatch repair and micro-RNA abnormalities compared to left-sided CRC [22,23].

In terms of the timing of metastasis, synchronous lesions were previously associated with poorer survival outcomes [5,21]. However, Bockhorn et al. and Mekenkamp’s studies demonstrated similar OS as well as disease-free survival for both synchronous and metachronous CRLM [19,24]. Another study by Quireze Junior et al. noted an even worse mean overall and 3-year survival in the metachronous group than in the synchronous group, but with similar recurrence-free survival [25]. This was akin to our findings in our present study. Metachronous metastasis as a negative predictor of OS may be attributed to prior exposure to chemotherapy, and thus the development of partial resistance, differences in tumor microbiology, and delays in metachronous metastasis detection [24,25].

Thermal ablation could also provide local control for CRLM cases with good survival outcomes, as seen in the present study and in previous reports. OS was comparable for patients who underwent either RFA or resection at 1 year (95.8–97.8% vs. 95.0–95.7%) and 3 years (66.8–69.8% vs. 60.1–71.6%). However, liver tumor progression-free survival has been reported to be shorter for RFA when tumors were more than 3 cm, but were found to be similar for smaller tumors [11,12]. For unresectable CRLM, combining RFA and systemic therapy was superior than systemic chemotherapy alone, having a 3-, 5-, and 8-year overall survival of 56.9% vs. 55.2%, 43.1% vs. 30.3%, and 35.9% vs. 8.9%, respectively [26]. Hepatic resection and RFA may likewise be combined for the additional control of multiple and bilobar liver metastases. Chiappa et al. reported considerably greater 5-year disease-free survival (50% vs. 33.9%) and 5-year overall survival (80% vs. 49%) for patients treated with both resection and RFA compared to resection alone of CRLM [27]. These findings are consistent with the results in our study. Our Kaplan–Meier curves as well as the computed median survival time and 5-year overall survival all demonstrated a definite survival benefit for patients with CRLM who underwent either surgical resection, RFA, or combination compared to systemic therapy alone or best supportive care.

Since data were yielded from a database retrospectively, certain limitations were present in this study. Details regarding the specific systemic therapy used, response to the treatment, recurrence-free survival, number, surgical method, tumor molecular characteristics, as well as complications from resection and RFA were not available for further analysis, which could possibly have an effect on the survival outcome. Furthermore, there were a limited number of RFA and combined cases included in this study.

## 5. Conclusions

In conclusion, aggressive local treatment by hepatic resection, RFA, or combined resection and RFA clearly provides a survival advantage over systemic therapy alone or best supportive care among patients with CRLM.

## Figures and Tables

**Figure 1 cancers-15-04434-f001:**
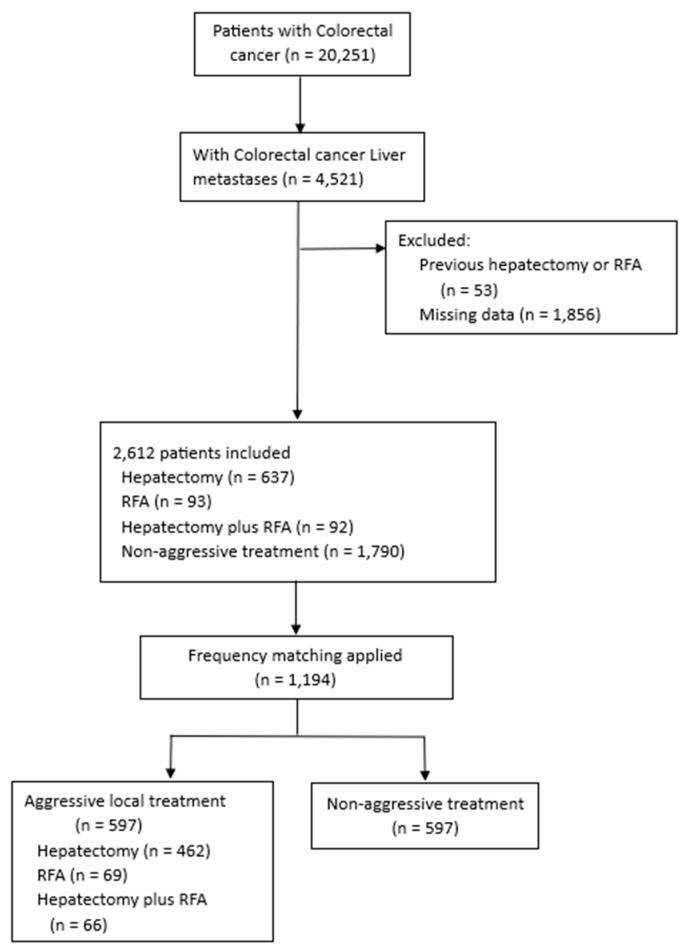
Flowchart of patient selection.

**Figure 2 cancers-15-04434-f002:**
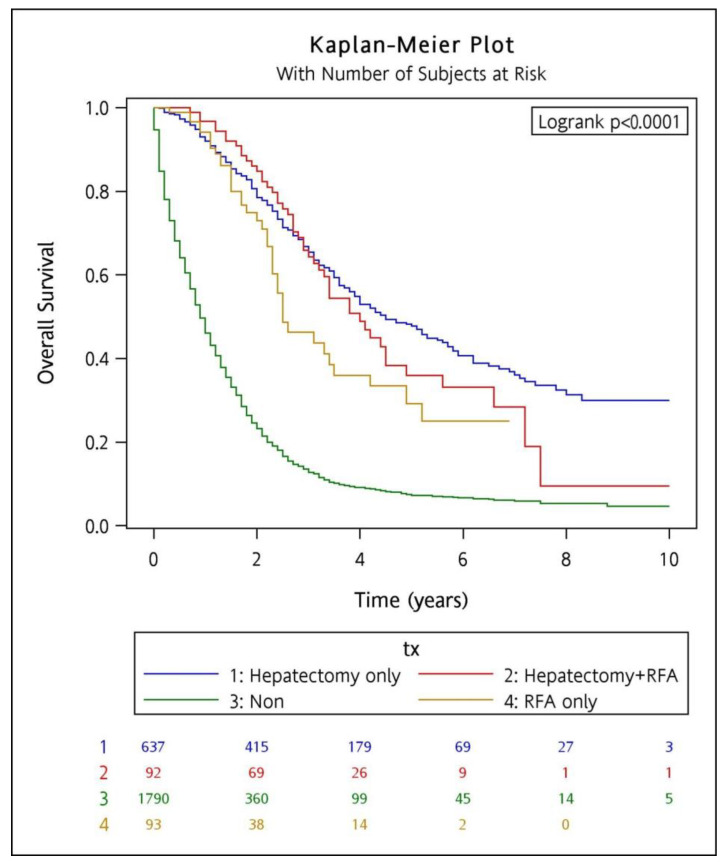
Kaplan–Meier curves for the entire cohort comparing the overall survival among the different treatment groups (tx). The numbers below denote the number of patients at risk in each group with a 2-year interval. Blue for those who received hepatectomy alone, red for combined hepatectomy and RFA, yellow for RFA alone, and green for non-aggressive treatment.

**Figure 3 cancers-15-04434-f003:**
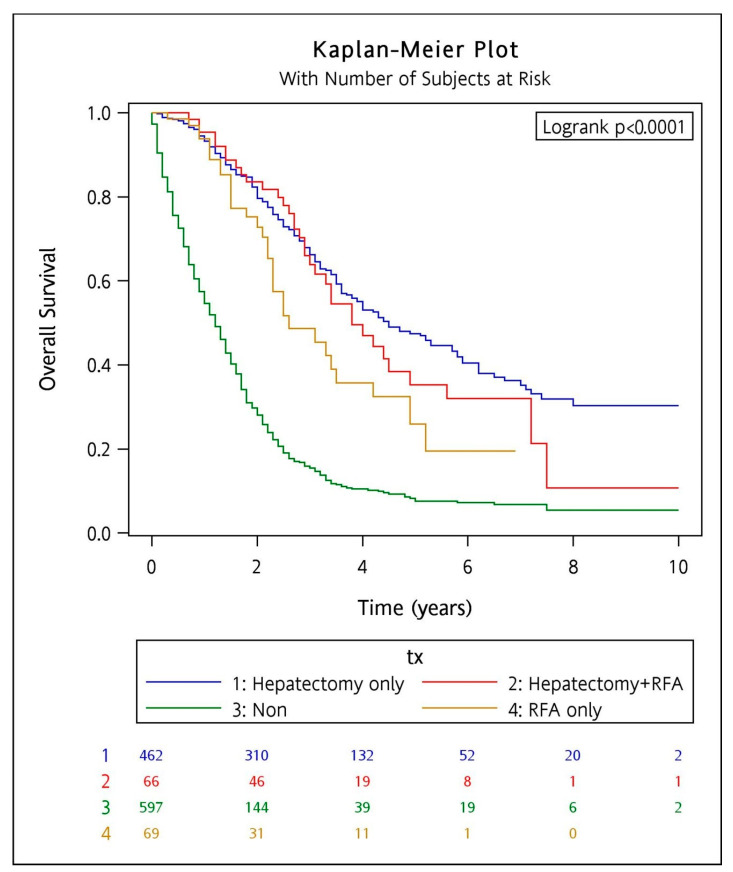
Kaplan–Meier curves for the frequency-matched cohort comparing the overall survival among the different treatment groups (tx). The numbers below denote the number of patients at risk in each group with a 2-year interval. Blue for those who received hepatectomy alone, red for combined hepatectomy and RFA, yellow for RFA alone, and green for non-aggressive treatment.

**Table 1 cancers-15-04434-t001:** Baseline characteristics of patients with colorectal cancer liver metastasis.

Variable	Hepatectomy + RFA		Hepatectomy		RFA		Non-Aggressive		*p*-Value
	*n*	%	*n*	%	*n*	%	*n*	%	
Total	92		637		93		1790		
Gender									0.757
Female	34	(37.0)	252	(39.6)	33	(35.5)	720	(40.2)	
Male	58	(63.0)	385	(60.4)	60	(64.5)	1070	(59.8)	
Age (years)									<0.001
<60	49	(53.3)	303	(47.6)	35	(37.6)	676	(37.8)	
≥60	43	(46.7)	334	(52.4)	58	(62.4)	1114	(62.2)	
Mean (SD)	59.6	(11.2)	60.6	(12.5)	64.3	(13.2)	63.9	(13.5)	<0.001
DM									0.147
No	67	(72.8)	515	(80.9)	68	(73.1)	1414	(79.0)	
Yes	25	(27.2)	122	(19.1)	25	(26.9)	376	(21.0)	
BMI (kg/m^2^), mean (SD)	23.9	(3.2)	24.4	(3.9)	24.5	(3.5)	23.3	(3.8)	<0.001
CEA (ng/mL), mean (SD)	109	(434)	120	(752)	53	(178)	432	(1846)	<0.001
Primary cancer site									0.011
Left	81	(88.0)	500	(78.5)	73	(78.5)	1339	(74.8)	
Right	11	(12.0)	137	(21.5)	20	(21.5)	451	(25.2)	
Clinical staging									<0.001
I	4	(4.4)	49	(7.7)	7	(7.5)	84	(4.7)	
II	7	(7.6)	74	(11.6)	8	(8.6)	146	(8.2)	
III	18	(19.6)	166	(26.1)	28	(30.1)	352	(19.7)	
IV	63	(68.5)	348	(54.6)	50	(53.8)	1208	(67.5)	
Clinical T staging									<0.001
1	3	(3.3)	5	(0.8)	1	(1.1)	18	(1.0)	
2	8	(8.7)	93	(14.6)	11	(11.8)	165	(9.2)	
3	51	(55.4)	350	(55.0)	56	(60.2)	857	(47.9)	
4	29	(31.5)	145	(22.8)	23	(24.7)	554	(31.0)	
Missing	1	(1.1)	44	(6.9)	2	(2.2)	196	(11.0)	
Clinical N staging									<0.001
0	20	(21.7)	178	(27.9)	21	(22.6)	365	(20.4)	
1	37	(40.2)	207	(32.5)	41	(44.1)	529	(29.6)	
2	30	(32.6)	205	(32.2)	27	(29.0)	665	(37.2)	
Missing	5	(5.4)	47	(7.4)	4	(4.3)	231	(129)	
Clinical M staging									<0.001
0	29	(31.5)	289	(45.4)	43	(46.2)	582	(32.5)	
1	63	(68.5)	348	(54.6)	50	(53.8)	1208	(67.5)	
Missing		(0)		(0.1)		(0)		(0.2)	
Liver metastases timing *									0.076
Synchronous	45	(48.9)	290	(45.5)	30	(32.3)	814	(45.5)	
Metachronous	47	(51.1)	347	(54.5)	63	(67.7)	976	(54.5)	
Extrahepatic metastasis									<0.001
No	71	(77.2)	503	(79.0)	62	(66.7)	787	(44.0)	
Yes	21	(22.8)	134	(21.0)	31	(33.3)	1003	(56.0)	
Systemic therapy									0.030
No	8	(8.7)	87	(13.7)	14	(15.0)	312	(17.4)	
Yes	84	(91.3)	550	(86.3)	79	(85.0)	1478	(82.6)	
Neoadjuvant (Pre HR)									<0.001
No	44	(47.8)	424	(66.6)	41	(44.1)	-	-	
Yes	48	(52.2)	213	(33.4)	52	(55.9)	-	-	
Adjuvant (Post HR)									<0.001
No	33	(35.9)	146	(22.9)	47	(50.5)	-	-	
Yes	59	(64.1)	491	(77.1)	46	(49.5)	-	-	

* Synchronous CRLM were defined as metastatic liver lesions found within 30 days of the primary diagnosis.

**Table 2 cancers-15-04434-t002:** Factors supporting overall survival for patients with colorectal cancer liver metastasis.

Variable	HR (95% CI)	*p*-Value	aHR (95% CI) *	*p*-Value
Sex				
Female	Reference		Reference	
Male	0.91 (0.82–0.99)	0.035	0.89 (0.81–0.97)	0.011
Age (years)				
<60	Reference		Reference	
≥60	1.31 (1.20–1.44)	<0.001	1.20 (1.09–1.32)	<0.001
DM				
No	Reference		Reference	
Yes	1.09 (0.97–1.22)	0.138	1.00 (0.89–1.12)	0.982
BMI (kg/m^2^)				
<25	Reference		Reference	
≥25	0.80 (0.72–0.88)	<0.001	0.92 (0.83–1.01)	0.080
CEA (ng/mL)				
≤5	Reference		Reference	
>5	1.84 (1.64–2.07)	<0.001	2.14 (1.89–2.42)	<0.001
Primary cancer site				
Left	Reference		Reference	
Right	1.40 (1.26–1.56)	<0.001	1.35 (1.22–1.51)	<0.001
Liver metastases timing				
Synchronous	Reference		Reference	
Metachronous	1.14 (1.04–1.24)	0.007	1.26 (1.14–1.39)	<0.001
Extrahepatic metastasis				
No	Reference		Reference	
Yes	2.10 (1.91–2.30)	<0.001	1.46 (1.33–1.60)	<0.001
Systemic therapy				
No	Reference		Reference	
Yes	0.71 (0.63–0.80)	<0.001	0.70 (0.62–0.79)	<0.001
Liver metastases treatment methods				
Hepatectomy + RFA	0.25 (0.19–0.34)	<0.001	0.28 (0.21–0.37)	<0.001
Hepatectomy only	0.22 (0.20–0.26)	<0.001	0.24 (0.21–0.27)	<0.001
RFA only	0.29 (0.29–0.41)	<0.001	0.33 (0.24–0.45)	<0.001
Non local therapy	Reference		Reference	

* Multivariable analysis was conducted by using Cox proportional hazards models that adjusted for sex, age, DM, CEA status, BMI status, primary cancer site, liver metastases timing, extrahepatic metastasis, chemotherapy, and liver metastases treatment methods.

**Table 3 cancers-15-04434-t003:** Frequency matching analysis of selected cases from hepatectomy and/or radiofrequency ablation vs. non-aggressive treatment groups (1:1).

Variable	Hepatectomy/RFA		Non-Aggressive		*p*-Value
	*n*	%	*n*	%	
Total	597		597		
Sex					1.00
Female	205	(34.3)	205	(34.3)	
Male	392	(65.7)	392	(65.7)	
Age (years)					1.00
<60	272	(45.6)	272	(45.6)	
≥60	325	(54.4)	325	(54.4)	
Mean (SD)	61.6	(11.5)	61.6	(11.5)	1.00
DM					0.512
No	475	(79.6)	484	(81.1)	
Yes	122	(20.4)	113	(18.9)	
BMI (kg/m^2^), mean (SD)	24.4	(3.7)	23.5	(3.9)	<0.001
CEA (ng/mL), mean (SD)	99	(490)	528	(2028)	<0.001
Primary cancer site					0.360
Left	475	(79.6)	462	(77.4)	
Right	122	(20.4)	135	(22.6)	
Clinical staging					0.006
I	35	(5.9)	31	(5.2)	
II	54	(9.1)	41	(6.9)	
III	148	(24.8)	108	(18.1)	
IV	360	(60.3)	417	(69.9)	
Clinical T staging					<0.001
1	6	(1.0)	5	(0.8)	
2	79	(13.2)	64	(10.7)	
3	336	(56.3)	293	(49.1)	
4	145	(24.3)	155	(26.0)	
Missing	31	(5.2)	80	(13.4)	
Clinical N staging					<0.001
0	146	(24.5)	127	(21.3)	
1	211	(35.3)	171	(28.6)	
2	202	(33.8)	210	(35.2)	
Missing	38	(6.4)	89	(14.9)	
Clinical M staging					<0.001
0	237	(39.7)	180	(30.2)	
1	360	(60.3)	417	(69.9)	
Missing	0	(0)	3	(0.2)	
Liver metastases timing *					1.00
Synchronous	301	(50.4)	301	(50.4)	
Metachronous	296	(49.6)	296	(49.6)	
Extrahepatic metastasis					1.00
No	431	(72.2)	431	(72.2)	
Yes	166	(27.8)	166	(27.8)	
Systemic therapy					1.00
No	56	(9.4)	56	(9.4)	
Yes	541	(90.6)	541	(90.6)	
Neoadjuvant (Pre HR)					
No	371	(62.1)	-	-	
Yes	226	(37.9)	-	-	
Adjuvant (Post HR)					
No	160	(26.8)	-	-	
Yes	437	(73.2)	-	-	
Treatment					
Hepatectomy + RFA	66	(11.1)	-	-	
Hepatectomy only	462	(77.4)	-	-	
RFA only	69	(11.6)	-	-	

* The data were matched by age, sex, liver metastases timing, extrahepatic metastasis, and chemotherapy.

**Table 4 cancers-15-04434-t004:** Risk of overall survival for colorectal cancer patients with liver metastasis.

Treatment	HR (95% CI) *	*p*-Value
Full data		
Hepatectomy + RFA	0.18 (0.08–0.39)	<0.001
Hepatectomy only	0.15 (0.11–0.21)	<0.001
RFA only	0.10 (0.04–0.25)	<0.001
Non	Reference	
Female		
Hepatectomy + RFA	0.32 (0.08–1.22)	0.095
Hepatectomy only	0.12 (0.06–0.22)	<0.001
RFA only	0.11 (0.02–0.56)	0.008
Non	Reference	
Male		
Hepatectomy + RFA	0.15 (0.06–0.38)	<0.001
Hepatectomy only	0.15 (0.10–0.22)	<0.001
RFA only	0.08 (0.02–0.26)	<0.001
Non	Reference	
Age < 60 years		
Hepatectomy + RFA	0.22 (0.08–0.59)	0.003
Hepatectomy only	0.19 (0.12–0.29)	<0.001
RFA only	0.08 (0.02–0.37)	0.001
Non	Reference	
Age ≥ 60 years		
Hepatectomy + RFA	0.12 (0.03–0.46)	0.002
Hepatectomy only	0.12 (0.07–0.19)	<0.001
RFA only	0.10 (0.03–0.36)	<0.001
Non	Reference	

* HR is calculated using stratified Cox model for matched data and adjusted by DM, CEA status, BMI status, and primary cancer site.

## Data Availability

Data are contained within the article.

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
