# Peer review of "Survival Evidence of Local Control for Colorectal Cancer Liver Metastases by Hepatectomy and/or Radiofrequency Ablation"

_cancers, 2023, doi:10.3390/cancers15184434_

Round 1
Reviewer 1 Report
This is a very nice study describing the impact of locally aggressive treatment versus no-aggresive therapy in patients with CRLM. The manuscript is well written and easy to read. The results are of importance since they clearly indicate a beneficial value of surgery/ ablative treatment to control disease in patients with CRC and hepatic metastasis, and thus, confirm the current state of art in intersisciplinary treatment approaches by using a large database. As mentioned by the authors, restrospective character of the study remains the major limiting factor. As demonstrtaed in Table 1, patients only receiving systemic therapies were older, more sick and were suffering from larger and more aggressive tumors. This is a very important selection bias, that could finally only be avoided by a prospective controlled trial, which however, may nowadays not be appropriate for ethical reasons in view of our recent clinical experiences and developments in hepatic surgery and systemic treatments etc. Nonetheless, despite existing limitations regarding study character, I recommend publication of the interesting manuscript, since it provides another good argue to intensively discuss options aof surgery and local ablation in each case of this specific subset.
There are some additional points that should be addressed by the authors:
1. Table 2 should not be titeld "risk for OS" but rather "factors supporting/promoting OS" for example.
3. The authors should provide more data on the extent of hepatectomey (major/minor/trisectorectomy, portal venous ligation, ALPPS etc) and RFA
2. Since study characteristics is rather heterogenous and non-aggressively treated patients seem to be more sick than thsoe of the other subgroups, OS is only one relevant aspect. The real value of aggressive intrahepatic surgery could be better described by recurrence-free survival/progression-free survival. If available, the authors should include respective data/analysis.
3. What were the contraindications for surgery/ablative treatment in the reference group: general/functional status? sarcopenia? liver dysfunction due to systemic treatment? fibrosis/steatosis/cirrhosis. Please give some more insights.
acceptable
Author Response
There are some additional points that should be addressed by the authors:
Point 1
- Table 2 should not be titeld "risk for OS" but rather "factors supporting/promoting OS" for example.
Response 1: Thank you very much for this suggestion! The title for table 2 is already revised.
Point 2
- The authors should provide more data on the extent of hepatectomey (major/minor/trisectorectomy, portal venous ligation, ALPPS etc) and RFA
Response 2: We are sorry that data about the details of surgical method such as extent of hepatectomy or ALPPS were not available from the incomplete medical records of database. We have acknowledged this study limitation in Discussion (page 12).
Point 3
- Since study characteristics is rather heterogeneous and non-aggressively treated patients seem to be more sick than those of the other subgroups, OS is only one relevant aspect. The real value of aggressive intrahepatic surgery could be better described by recurrence-free survival/progression-free survival. If available, the authors should include respective data/analysis.
Response 3: Thank you for the suggestion! We, however do not have available data regarding the progression-free survival or recurrence-free survival. We have acknowledged this study limitation in Discussion (page 12).
Point 4
- What were the contraindications for surgery/ablative treatment in the reference group: general/functional status? sarcopenia? liver dysfunction due to systemic treatment? fibrosis/steatosis/cirrhosis. Please give some more insights.
Response 4: Thank you for your comments. These patients were discussed in a multi-disciplinary meeting to recommend the procedure precedence. As shown in table 1, the non-aggressive treatment group had a higher ratio of extrahepatic metastasis compared to the local control treatment groups, suggesting that extrahepatic metastasis is the primary reason why the reference group did not receive surgery nor RFA. However, our frequency matching analysis including extrahepatic metastasis in the matching process, showed the definite survival advantage of local control with resection, RFA or combination even though the patients had extrahepatic metastasis. We have discussed more for this issue in Discussion (page 11).

Reviewer 2 Report
The paper by Canseco et all on outcomes following surgery or RFA for local control CLRM is a comprehensive review of a large multi-centre database
Overall the paper is well-written, concise and reports on a topical issue that is increasingly becoming an issue across the globe. The benefits of multi-modal tx for CRC and survival benefits for those with CLRM in recent decades has meant that management strategies are becoming more radical and complex.
Overall the paper needs some revision - See below
Congrats on your work
Suggestions /edits:
1. Abstract:
- Worth providing actual difference ion survival in terms of % OVS or median survival in differences
2. Intro:
- Worth highlighting - In terms of surgical - Re: Margins and close margins versus R2
- Be specific has to why ablation can be utilized
-Size cutoff for ablation
3. Methods:
-Why missing data on 1800 pts - More specifics....Could some of these not be include in some aspects of the review ?
-How many hospitals in the CGMH group - What is their make-up......Academic units, regional, specialist etc?
-The term synchronous and using a 30 day definition would be at odds with most definitions by others parts of world - Look at the recent publication: HPB (Oxford)
doi: 10.1016/j.hpb.2023.05.360.
- Provide more detail on your frequency matching - Manually or systemically
4. Results
- Did you have molecular details - KRAS, BRAF etc ?...If not why not - These behave as very different tumors
- Please report median and 5-yrs OVS in months and % respectively throughout
- What about the impact that complication had on survival - Have you data on this - Very relevant to include especially when discussing complex management strategies
5. Discussion
- Limitations needs to be re-considered more....No data on complications, molecular status, distribution and timing of CLRM, repose to chemo
6. Figures/Tables/Refs
- No major concerns
Small grammatical issues
Shortened sentences in results section - to be clear and concise
Author Response
Point 1:
- Abstract: - Worth providing actual difference ion survival in terms of % OVS or median survival in differences
Response 1: Thank you very much for your kind suggestion. We have increased the data of 5-year survival in abstract. "The 5-year overall survival (OS) was as follows: hepatectomy alone 47.8%, combined hepatectomy plus RFA 35.9%, and RFA alone 29.2% versus non-aggressive treatment group 7.4%".
- Intro:
Point 2
- Worth highlighting - In terms of surgical - Re: Margins and close margins versus R2
Response 2: Thank you very much for your valuable comment. We have increase the content about margins in Introduction (page 2). "Ideally, R0 resection should be achieved whether surgery is done as curative or palliative treatment. In a study by Park et al, palliative resection for metastatic colorectal cancer with negative margin (R0) showed a significantly longer survival compared to patients with either positive margin (R1) or grossly residual tumor (R2) (51.3 months versus 19.1 months) and those without resection (14.1 months)." A new reference (ref. 9) has been added.
Reference: Park, JH, Kim, TY, Lee KH, Han SW, Oh DY, Im SA, Kang GH, Chie EK, Ha SW, Jeong SY, et al. The beneficial effect of palliative resection in metastatic colorectal cancer. Br J Cancer. 2013; 108: 1425-1431.
Point 3
- Be specific has to why ablation can be utilized
Response 3: We have increased the content why ablation can be utilized in Introduction (page 2). "Thermal ablation is a less invasive procedure that has potential to provide curative intent, as an alternative to resection. This is especially beneficial for patients with unresectable CRLM, patients with co-morbidities preventing them from surgery, or those with insufficient liver reserve."
Point 4
-Size cutoff for ablation
Response 4: We have increased the content about size cutoff for ablation in Introduction (page 2). "The consensus made by an expert panel recommended tumors less than 3 cm as the preferred size for thermal ablation but distinct tumors <5 cm may also have good thermal ablation outcomes depending on their location and the ablation method used."
- Methods:
Point 5
-Why missing data on 1800 pts - More specifics....Could some of these not be include in some aspects of the review ?
Response 5: Missing data is an inevitable problem when conducting such hospital-based analysis based on retrospective reviews of medical charts. In this study, we excluded the patients with 3 more missing data. However, multivariate analysis using Cox proportional hazards excluded observations with any missing variables.
Point 6
-How many hospitals in the CGMH group - What is their make-up......Academic units, regional, specialist etc?
Response 6: We have increased the content about CGMH group in Method (page 3). "CGMH is the largest medical care system in Taiwan, consisting of 4 tertiary‐care medical centers and 3 major teaching hospitals. This medical care system provides nearly 10% of medical service used by the Taiwanese people annually, with more than 10,000 beds and over 280,000 inpatients per year."
Point 7
-The term synchronous and using a 30 day definition would be at odds with most definitions by others parts of world - Look at the recent publication: HPB (Oxford) 2023 Jul 13;S1365-182X(23)00514-2. doi: 10.1016/j.hpb.2023.05.360.
Response 7: Thank you very much for your valuable comment. The definition of synchronous CRLM varies in the literature in regards to the timing of liver disease diagnosis in relationship to the diagnosis of the primary tumor, ranging from 0 all the way to 12 months. Most studies recommend that "early metachronous metastases" applies to those absent at presentation but detected within 12 months of diagnosis of the primary tumor and late metachronous metastases applied to those detected after 12 months, as your suggesting reference (HPB (Oxford) 2023 Jul 13;S1365-182X(23)00514-2). However, most of the definitions of synchronous CRLM include liver metastasis detected at or before diagnosis or surgery of the primary tumor, but there are some others who include metastases detected up to 1 month, 3 months, 4 months or 6 months following diagnosis (Adam R, et al. Cancer Treat Rev. 2015; 41: 729-41).
Point 8
- Provide more detail on your frequency matching - Manually or systemically
Response 8: We have explained more details for frequency matching in Method (page 5). "The data were obtained by systematically conducting simple randomization sampling with frequency matching by age, gender, liver metastases timing, extrahepatic metastasis, and systemic therapy."
- Results
Point 9
- Did you have molecular details - KRAS, BRAF etc ?...If not why not - These behave as very different tumors
Response 9: We totally agree that the molecular factors of the tumors affect prognosis among colorectal cancer patients. However, these information were only available in recent 5 years, whereas our data were collected from January 2004 and December 2017 thus ours were limited.
Point 10
- Please report median and 5-yrs OVS in months and % respectively throughout
Response 10: We have increased the data of 5 year OS and median survival time in Result (page 6). "The median survival time for hepatectomy, combined hepatectomy plus RFA and RFA alone (54 months, 48 months and 30 months respectively), were significantly longer compared to the non-aggressive treatment group (10.8 months). A similar trend was likewise seen for the 5-year OS of hepatectomy, combined hepatectomy plus RFA and RFA alone (47.8%, 35.9% and 29.2% respectively) in contrast to the non-aggressive treatment group with 7.4%." Also, in page 8 " After frequency matching, both the median survival time and 5-year OS remained significantly higher in the local control treatment groups compared to the non-aggressive treatment group (combined hepatectomy plus RFA 45.6 months and 35.3%, hepatectomy alone 54.0 months and 47.5%, RFA alone 31.2 months and 26% versus non-aggressive treatment group 14.4 months and 8.3%)."
Point 11
- What about the impact that complication had on survival - Have you data on this - Very relevant to include especially when discussing complex management strategies
Response 11: We are sorry that data about the complications were not available from the incomplete patient records of database. We have acknowledged this study limitation in Discussion (page 12).
- Discussion
Point 12
- Limitations needs to be re-considered more....No data on complications, molecular status, distribution and timing of CLRM, repose to chemo
Response 12. We do understand that our study has several limitations due to its retrospective nature and because we were dependent on the available information from the CGMH database. Will mention all of these in the limitations section of our paper (page 12). Thank you!
Comments on the Quality of English Language
Small grammatical issues
Point 13
Shortened sentences in results section - to be clear and concise
Response 13. Thank you for your suggestion! We will improve on sentence construction.

Reviewer 3 Report
This is a retrospective review on colorectal cancer liver metastases with focus on local control and how it impacts survival. This is a topic with extensive publications already present. The strength of the study is the large number of patients 2,612 however the study has significant limitations
1) there is no data on local recurrence free survival which is most important when comparing local therapies
2) most academic programs have switched to MWA so the study is not necessarily applicable to many countries
3) few data are given about what systemic therapies were used, how many lines, what constitutes non aggressive local therapy etc
4) No selection criteria for hepatectomy vs RFA are given (size location etc)
overall minor revisions are recommended
Author Response
1) there is no data on local recurrence free survival which is most important when comparing local therapies
Response 1: Thank you very much for the suggestion! The main objective of our paper was to compare the overall survival of the CRLM patients who received local control treatment (either with resection, RFA or combined surgery and RFA) versus those who did not undergo aggressive treatment. The emphasis was not in comparison of surgery versus RFA or combination. However, we do not have available data regarding the local recurrence-free survival. We have acknowledged this study limitation in Discussion (page 12).
2) most academic programs have switched to MWA so the study is not necessarily applicable to many countries
Response 2: Both RFA and MWA have their own merits and limitations in treating liver tumors. Several studies on RFA and MWA have shown comparable results in terms of efficacy and safety. In our institution, both RFA and MWA are available but the later was available in recent 5 years, whereas our data were collected from January 2004 and December 2017 thus data about MWA were limited for analysis.
3) few data are given about what systemic therapies were used, how many lines, what constitutes non aggressive local therapy etc
Response 3: The non-aggressive treatment group consists of patients who either received systemic therapy alone or best supportive care. We apologize that further details regarding the exact systemic treatments given were not available from the database.
4) No selection criteria for hepatectomy vs RFA are given (size location etc)
Response 4: The patients who underwent surgery and/ or RFA were deemed resectable and/or ablatable by the multi-disciplinary team who discussed the cases. We have increased the content about size cutoff for ablation in Introduction (page 2). "The consensus made by an expert panel recommended tumors less than 3 cm as the preferred size for thermal ablation but distinct tumors <5 cm may also have good thermal ablation outcomes depending on their location and the ablation method used."

Reviewer 4 Report
The study by Canseco et al. performed a statistical analysis of the survival rate of patients with colorectal cancer liver metastasis after hepatectomy and/or radiofrequency ablation. This analysis is overall informative, however, several points need to be addressed to make sure the conclusion is solid.
1. The survival curve of patients with hepatectomy plus RFA is very similar to patients with non-local therapy at the time of about 8 years in Figure 2, this is very concerning regarding the authors' central conclusion of the whole manuscript. Also, the survival curve ends about 7 years with RFA only group, but not the other groups, is there a specific reason for that? This is a similar question to Figure 3.
2. Legends for Figures 2 and 3 were missing.
English is fine.
Author Response
Point 1.
- The survival curve of patients with hepatectomy plus RFA is very similar to patients with non-local therapy at the time of about 8 years in Figure 2, this is very concerning regarding the authors' central conclusion of the whole manuscript. Also, the survival curve ends about 7 years with RFA only group, but not the other groups, is there a specific reason for that? This is a similar question to Figure 3.
Response 1: Thank you very much for your valuable comment. The median OS in patients with hepatectomy plus RFA was 48 months. Only one patient had a follow-up longer than 8 years, thus small case number would lead to sudden decrease in OS after 8 years of follow-up. On the other hand, there were fewer cases of RFA-only patients, and the longest follow-up period to the study end date was up to 7 years in this subgroup population.
- Legends for Figures 2 and 3 were missing.
Response 2: The legends are found under the Kaplan-Meier curves. Blue line is for the hepatectomy only group, red line is for the combined hepatectomy plus RFA, yellow line is for the RFA only and green line is for the non-aggressive treatment group.

Round 2
Reviewer 4 Report
The authors addressed my comments, for point #2, I was asking for legend explaining the numbers underneath the colored numbers as well as annotating all the abbreviations such as "tx" in the box, please make sure to include these information in the legend, as the current legend is insufficient at all.
Fine.
Author Response
The authors addressed my comments, for point #2, I was asking for legend explaining the numbers underneath the colored numbers as well as annotating all the abbreviations such as "tx" in the box, please make sure to include these information in the legend, as the current legend is insufficient at all.
Response: Thank you very much for your comment. The legends at figure 2 and 3 are revised. "Kaplan-Meier curves for the entire cohort comparing the overall survival among the different treatment groups (tx). The numbers below denote the number of patients at risk in each group with 2-year interval. Blue for those who received hepatectomy alone, Red for combined hepatectomy and RFA, Yellow for RFA alone and Green for non-aggressive treatment."
